

# Wavelength calibration of Brewer spectrophotometer using a tuneable pulsed laser and implications to the Brewer ozone retrieval

Alberto Redondas[1,2], Saulius Nevas[3], Alberto Berjón[4,2], Meelis-Mait Sildoja[3], Sergio Fabian León-Luis[1,2], Virgilio Carreño[1,2], and Daniel Santana[4,2]

[1]Agencia Estatal de Meteorología, Izaña Atmospheric Research Center, Spain
[2]Regional Brewer Calibration Center for Europe, Izaña Atmospheric Research Center, Tenerife, Spain
[3]Physikalisch-Technische Bundesanstalt (PTB), Braunschweig, Germany
[4]University of La Laguna, Department of Industrial Engineering, S.C. de Tenerife, Spain

*Correspondence to:* Alberto Redondas (aredondasm@aemet.es)

**Abstract.**

In this contribution we present the wavelength calibration of the traveling reference Brewer spectrometer of the Regional Brewer Calibration Center for Europe (RBCC-E) at PTB in Braunschweig, Germany. The wavelength calibration is needed for the calculation of the ozone absorption coefficients used by the Brewer ozone algorithm. In order to validate the standard

procedure for determining Brewer's wavelength scale, a calibration has been performed by using a tuneable laser source at PTB in the framework of the EMRP project ENV59 ATMOZ "Traceability for the total column ozone". Here we compare these results to those of the standard procedure for the wavelength calibration of the Brewer instrument. Such a comparison allows validating the standard methodology used for measuring the ozone absorption coefficient with respect to several assumptions. The results of the laser-based calibrations reproduces those obtained by the standard operational methodology and shows that

there is a underestimation of 0.8% due the use of the parametrized slit functions.

## 1 Background

Nowadays the primary ground-based instruments used to report total column ozone (TOC) are Dobson and Brewer Spectrophotometers. These instruments make measurements of the UV irradiances, and through a well-defined process a TOC value is

produced. The Brewer spectrometer (Brewer, 1973; Kerr et al., 1981; Kerr, 2010) was introduced in the 1980s as an automatic device measuring direct solar UV radiation and global UV irradiance. Both, the Brewer and the Dobson instruments were considered by the World Meteorological Organization (WMO) in the framework of the Global Atmosphere Watch program (GAW) as the standard instruments for TOC monitoring. The wavelength calibration is needed for calculation of the ozone absorption coefficient used by the Brewer ozone retrieval algorithm. The Brewer spectrophotometer has two operating modes.

In the ozone mode, used for the total ozone column measurements, the diffraction grating stays at a fixed position while the





six operational wavelengths are selected by a rotating slit mask. The scanning mode is used to perform spectral irradiance measurements in the ultraviolet (UV) spectral range. In this mode, the slits are fixed and the spectral scan is carried out by turning the diffraction grating. The usual wavelength calibration procedure is performed in the scanning mode by analyzing recorded emission lines of the spectral discharge lamps, which are usually mercury (Hg), cadmium (Cd) , and zinc (Zn). Using the spectral lines provided in Table 1 allow us to determine the central wavelengths and the corresponding FWHM (full width at half maximum) of the slit functions as well as the relation between the positions of the grating and the corresponding instrument wavelengths (dispersion relation) required to determine the operational wavelengths used for the ozone determination. To obtain the ozone absorption coefficients, the instrumental slit functions are convolved whit the Bass & Paur ozone absorption cross-section data. The use of the of the tuneable laser source allow us to:

– Calculate the ozone absorption coefcients from the calibration directly in the ozone mode. The normal determination of the ozone absorption coefficients involves scanning of the spectral lines in the scanning mode of the instrument so that the dispersion relation is required to convert the grating positions in micrometer steps to the respective wavelengths. Here we can determine the instrumental slit functions directly in the ozone mode by scanning them with the tuneable laser and weight them with the ozone absorption cross-sections without the need for the assumptions about the slit functions and the dispersion relations used in the normal calibration procedure.

– Calculate the dispersion relation based on regularly spaced reference spectral lines provided by the tuneable laser instead of the irregularly distributed emission lines of the Hg, Cd and Zn spectral lamps.

During the experiment we performed three measurements:

1. The standard method for the dispersion measurements using spectral lamps described in section 2.

2. Direct dispersion measurements (laser scanning).

   While the Brewer is measuring in the ozone mode and in the aerosol mode, the laser wavelength is swept ±2nm with a step of 0.04nm around the six Brewer operational wavelengths selected by the rotating slit mask for different grating positions (Figure 1)

3. Dispersion measurements using the tunable laser (Brewer scanning). While the laser is emitting at a fixed wavelength, the Brewer scans ±2nm around this wavelength in the scanning mode by moving the grating and using the 6 slits. Such Brewer scans are carried out at wavelengths ranging from 290 nm to 365 nm with an increment of 5nm. The results allow us to estimate the dispersion approximation error due to the lack of spectral lines at the end of spectral range of the Brewer spectrophotometer and due to the fact that the emission lines of the used lamps are not equally spaced.

## 2   Calibration of the Brewer spectrophotometer

The Brewer instrument measures the irradiance of direct sunlight at six nominal wavelengths ($\lambda$) in the UV range (303.2, 306.3, 310.1, 313.5, 316.8, and 320.1 nm), each spectral band covering a bandwidth of 0.5 nm (resolving power $\lambda/\delta\lambda$ of





**Table 1.** Emission lines of the discharge lamps used for Brewer calibration

| Lamp | Line (nm) | Slits |
| --- | --- | --- |
| Mercury (Hg) | 289.36 | 0–1 |
| Hg | 296.728 | 0–3 |
| Zinc (Zn) | 301.836 | 0–5 |
| Zn | 303.578 | 0–5 |
| Cd (multiplet) | 308.082 | 0-5 |
| Cd | 313.3167 | 0–5 |
| Cd | 326.1055 | 0–5 |
| Zn | 328.233 | 0–5 |
| Hg | 334.148 | 0–5 |
| Cd | 340.3652 | 0–5 |
| Cd | 349.995 | 4–5 |
| Cd (multiplet) | 361.163 | 5 |

around 600). The spectral resolution is achieved by a holographic grating in combination with a slit mask that selects the channel to be analyzed by a photomultiplier tube (PMT). The longest four wavelengths are used for the ozone calculation. Based on the Lambert-Beer law, the total ozone column in the Brewer algorithm can be expressed as:

$$X = \frac{F - ETC}{\alpha\mu} \tag{1}$$

5    where $F$ are the measured double ratios corrected for Rayleigh effects, $\alpha$ is the ozone absorption coefficient, $\mu$ is the ozone air mass factor, and $ETC$ is the extra-terrestrial constant. The $F$, $\alpha$ and $ETC$ parameters are weighted functions at the operational wavelengths with weighting coefficients $w$:

$$F = \sum_i^4 w_i F_i - \frac{p}{p_0}\beta_i\mu \tag{2}$$

$$\alpha = \sum_i^4 w_i \alpha_i \tag{3}$$

10  $$ETC = \sum_i^4 w_i F_{0i} \tag{4}$$





where, $\beta_i$ are the Rayleigh coefficients, $p$ is the climatological pressure at the measurement site, $p_0$ is the pressure at sea level, and $F_0$ are the individual extra-terrestrial constants at each wavelength. The weights $w = [1, -0.5, -2.2, 1.7]$ have been chosen so as to minimize the influence of $SO_2$ and verify:

$$\sum_i^4 w_i = 0 \qquad (5)$$

$$\sum_i^4 w_i \lambda_i = 0 \qquad (6)$$

This widely eliminates absorption features which depend, in local approximation, linearly on the wavelength, like for example the contribution from aerosols.

We can divide the calibration in three steps including instrumental calibration, wavelength calibration, and ETC transfer:

1. The instrumental calibration includes all the parameters that affect the measured signal counts ($F$), in particular, photo-multiplier dead time correction, temperature coefficients and filter attenuation.

2. Wavelength calibration is needed to determine the ozone absorption coefficient. The so-called "dispersion test" is used to obtain the particular wavelengths for the instrument and the slits, or instrumental functions, of each spectrophotometer. Note that the precise wavelengths of every Brewer spectrophotometer are slightly different from instrument to instrument.

3. Finally, the ETC transfer is performed by comparison with the reference Brewer instrument or, in the case of the reference instruments, by the Langley method.

The Brewer wavelength calibration follows the operative procedure (Gröbner et al., 1998; Kerr, 2002) used by the Regional Brewer Calibration Center-Europe (RBCC-E) at the calibration campaigns. In summary, the individual wavelengths (bands) in the Brewer instrument are selected through the use of a stainless steel mask of seven slits located at the focal plane of the spectrometer. The particular wavelength is determined by analyzing the measurements of a series of discharge lamps during so-caled dispersion test, which determines the central wavelength and FWHM of every slit. Then the wavelength setting is optimized to minimize the effect of wavelength shift during the operation of the instrument by performing the so-called sun-scan test (Lamb, K and Asbridge, A.I. , 1996). Finally, the ozone absorption coefficient is determined for every slit.

The ozone absorption coefficient were defined as (Vanier and Wardle, 1969; Bernhard et al., 2005; Wardle, 2008):

$$\widetilde{\alpha}(X, \mu) = \sum w_i \frac{\int \alpha(\lambda) * S(\lambda, \lambda') * F(\lambda, \lambda', X, \mu) d\lambda'}{\int S(\lambda, \lambda') * F(\lambda, \lambda', X, \mu) d\lambda'} \qquad (7)$$

Where $S$ is the instrumental slit function for the corresponding wavelength, $F$ is the sun spectrum that depends mostly on the ozone concentration and airmass, and $\sigma$ is the ozone cross-section at the temperature of $-46.3$ °C for Dobson and at $-45$ °C for Brewer instruments.

The Brewer operative method uses the following assumptions:





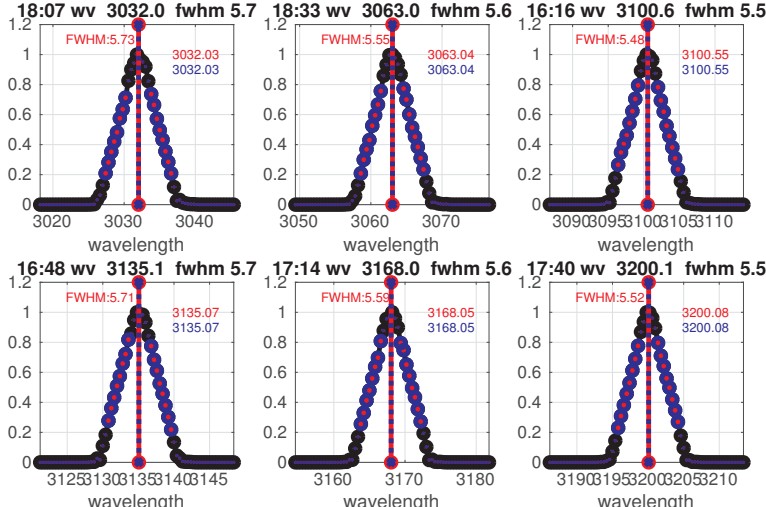

**Figure 1.** Results of Brewer measurements in the ozone mode obtained while the laser wavelength was changed every 0.04 nm . The central wavelength and the FWHM calculated are displayed in red using the same methodology of the dispersion test

1. Use "ideal" slits; the slit functions are parametrized as trapezoids, i.e., isosceles triangles truncated at 0.87 height.

2. Stray light is not considered, i.e., zero slit function values are assumed outside the triangle.

3. The FWHM of the triangle is dependent of the slit and it is derived from the dispersion test.

4. The ozone cross sections are expressed by the Bass & Paur absorption coefficient data set.

5   5. Solar spectrum is not considered , ($F==1$)

Under these assumptions, the ozone effective absorption is essentially obtained the same way as in the approximation method of Bernhard et al. (2005) used with Dobson spectrophotometers. (see Eq. 8).

$$\alpha_i = \frac{\int \sigma(\lambda) S_i(\lambda) \mathrm{d}\lambda}{\int S_i(\lambda) \mathrm{d}\lambda} \tag{8}$$

## 2.1 Dispersion Test

10   The Brewer spectrophotometer is constructed based either on a single or a double monochromator of modified Ebert–Fastie type, generally referered to as single or double Brewer, respectively. The first monochromator disperses the incoming radiation onto six exit slits. In the case of the double Brewer , the six exit slits (intermediate slits) of the first monochromator are the entrance slits to a second monochromator that is used in subtractive mode. The wavelength is selected by choosing one of the





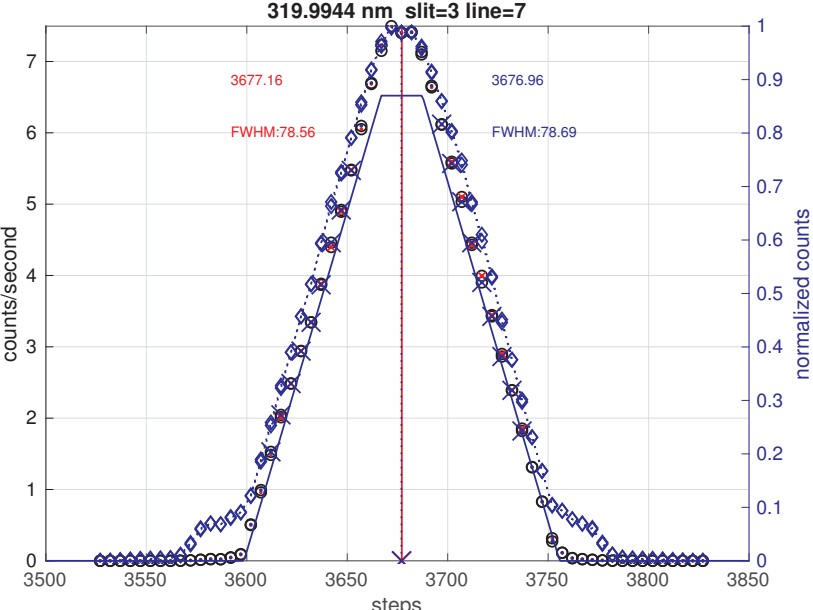

**Figure 2.** Slit function of the slit #3 (320 nm) measured in the scanning mode: the standard method uses the normalized values only between 0.2 and 0.8 (points with crosses). The figure shows the signals recorded during the scans (blue diamonds) and the non-linearity-corrected values (black circles) in counts/second. The hysteresis is evident from the asymmetry of the uncorrected signals in the low signal region of the plot. The center steps and the FWHMs calculated from the up-scan are shown in red while the values derived from the down-scanned slit function are presented in blue text. The central wavelength determination is not affected by the nonlinearity but the apparent FWHMs would be larger if the non-linearity correction were not applied. The resulting parametrized slits are represented on the right axis.

six exit slit (ozone mode) or rotating the grating (scanning mode). The rotation of the grating is managed by a drive mechanism consisting of a motor-driven micrometer linked to an arm that rotates the grating. The smallest wavelength increment corresponding to one stepper motor step varies steadily from approximately 8.0 pm to 7.0 pm ( 0.0080 nm ) Gröbner et al. (1998).

5   The dispersion relation, which provides the relation between the micrometer steps and the monochromator set wavelengths, is determined by scanning the emission lines as described in section 1. The line scans are carried out with an increment of 10 motor steps ( 0.7A). From the results, the central position and the FWHM of the slit function are calculated in motor steps assuming an isosceles triangle. The both sides of the peak are fitted to a straight line taking only the function values above 20% and bellow 80% of the normalized peak. The central point is calculated by the intersection point and the FWHM is the

10  width of the triangle (Figure 2 ). Finally, the dispersion relation is calculated using a quadratic polynomial or the cubic spline approximation. This relation is used to transform the previously determined central positions and FWHMs of the slit functions in micrometer steps to a wavelength scale.





The cubic approximation method of Gröbner et al. use knowledge of the optical design of the Brewer to transfer results of the spectral line measurements from one slit to the other slits, which reduces the number of free parameters to be adjusted compared with the quadratic method. However, there is a systematic difference between the ozone absorption coefcients calculated by the two methods. Both methods generally agree only in the ozone spectral range of the Brewer instrument (Redondas and
5 Rodriguez-Franco, 2012).

The stability of the wavelength calibration during Brewer operations is checked by measuring the internal Hg lamp. In most of the Brewers, the 302 nm double line (302.150 nm and 302.347 nm) is used due to its proximity to the Brewer operational wavelengths. However, for Brewer #185 and for an increasing number of other Brewers the test is performed using the more powerfull 296.7 nm line. The wavelength test includes 12 measurements of the line from the mercury lamp on slit 0, with
10 an increment of $\pm 10$ steps of the micrometer motor. The obtained curve for the line peak is compared to a stored reference one. The comparison is done by shifting the two scaned curves against each other and calculating the correlation coefficient between the two after each shift. The interpolated step number yielding the maximum of the correlation coefficient provides the reference micrometer position (Savastiouk, 2005). If the required adjustment of the micrometer position is more than one and a half motor steps, the test is repeated. Hence, the accuracy of the wavelength setting of the Brewer instrument achieved by
15 such an approach is defined by a rectangular probability distribution function on the interval of +-1.5 steps. Thus, the respective standard uncertainty is 1.5/sqrt(3)= 0.866 steps. This affects the ozone abortion coefficient by approximately of 0.10 $\mathrm{atmcm}^{-1}$ and the ozone concentration by 0.3%.

"

## 3 Pulsed laser-based measurements

### 3.1 Instrumental setup

For scanning the bandpass functions of the Brewer instrument, an upgraded PLACOS setup (Nevas et al., 2009) featuring a tuneable pulsed laser system based on an Optical Parametric Oscillator (OPO) was used at Physikalisch-Technische Bundesanstalt (PTB) in Braunschweig (Figure 3). The new OPO system generates 3-6 ns pulses at 1 kHz repetition rate in the spectral range from 210 nm to 2600 nm. The laser wavelengths were monitored during the measurements by a wavemeter
(laser spectrum analyzer) and a high-resolution spectrometer with an uncertainty of 0.01 nm. The laser beam was guided into the direct port of the Brewer spectrophotometer by using a liquid light guide. A fraction of the beam was directed to a monitor photodiode in order to account for the output power changes of the laser beam. The photocurrent of the silicon photodiode was measured by a charge meter.

In contrast with the standard calibration procedure, where the Brewer instrument scans the lines of the spectral lamps, in
this experiment the Brewer measures in ozone mode. Here, the Brewer grating is fixed at the ozone position while the coupled laser beam is measured using the seven slits (slit #1 is used to obtain the dark signal values). During these measurements the wavelength of the OPO system is scanned with 0.04 nm step. The experiment is complemented by the measurements in the





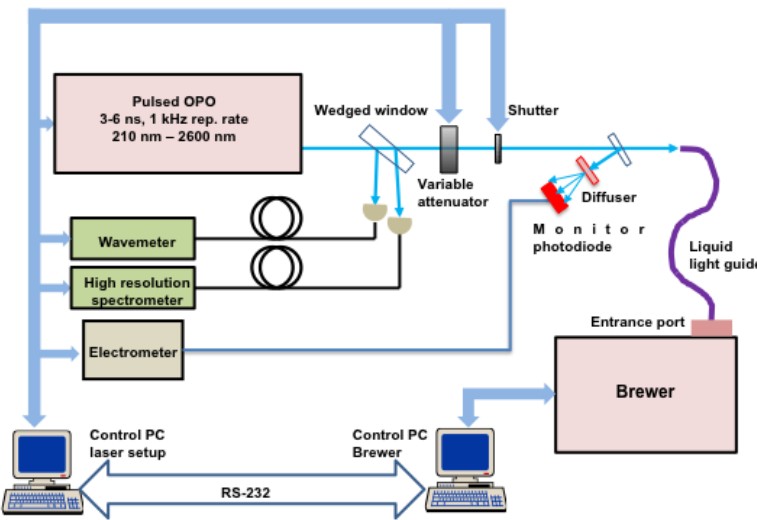

**Figure 3.** Pulsed optical parametric oscilator (OPO)-based setup at PTB that was used for measuring the slit functions of the Brewer spectrophotometer.

Brewer scanning mode, where the tuneable laser is used as a source of spectral lines covering the range from 290 nm to 360 nm on a regular grid with 5 nm step.

### 3.1.1 Non-linearity of the PMT

The Brewer detector system, which is based on a PMT, responds non-linearly to pulsed sources. For the measurements of
pulsed sources, the PMT manual advises to change the electronics configuration. As the main objective was to validate the operational wavelength calibration of the Brewer, we decided to keep the instrument configuration equivalent to that during the field operations. The non-linearity problem was solved by determining the respective correction function. For this purpose, the power of the laser beam was varied while simultaneously measuring signals of the PMT and the linear monitor photodiode.

     The ratios of the measured Brewer counts to the recorded monitor photodiode signals are shown in figure 4. The non-linearity
is evident in the figure together with hysteresis region near $10^4$ Brewer counts/seconds. The correction is not reliable around $10^3$ counts/seconds and lower than 100 counts/seconds. As we can control the power of the laser beam, it is possible to work on the "flat regions" of the non-linearity characteristics and apply the determined correction. This correction does not affect the calculated central wavelength of the slit functions, though, it does affect the determined FWHM values (Figure 2) if the correction is not applied.

We observed that the recorder dark signal values (measurements performed with the blocked slit #1) were highest immediately after exposing the PMT to the laser light. The dark signal of the PMT was then gradually fast decreasing with time after




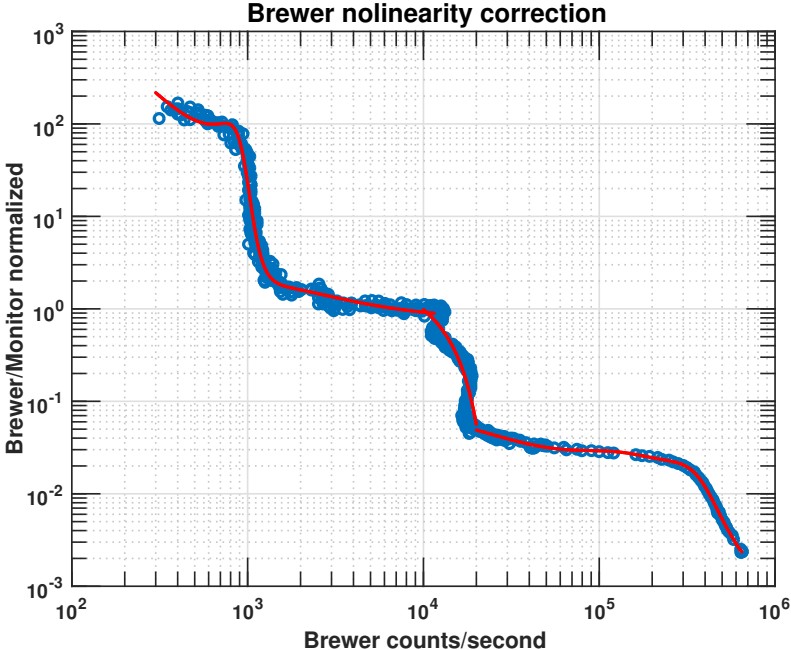

**Figure 4.** Log-log plot of the normalized ratio of measured Brewer counts to the monitor photodiode signal, which is proportional to the laser power, plotted as a function of the Brewer counts/s. The black points shows the measured points while the red curve is the fit used to correct the Brewer signals for the nonlinearity.

the excitation, which may cause the signal values obtained for slit #1 (measured immediately after slit #0) be higher than for the other slits measured afterwards.

### 3.1.2 Slit parametrization

The Brewer algorithms assume trapezoidal slits functions cut at 0.87 of the height (Figure 6) with the center wavelength and the FWHM calculated for every slit from the dispersion relation. The laser measurements allow us to estimate the effect on the ozone calculation if we use the directly measured slit functions instead of the parametrised ones. For this purpose we calculate the ozone absorption coefficients for the four ozone cross sections evaluated in the "ACSO" comitee ("Absorption Cross Sections of Ozone") ( Orphal et al. (2016).

Among the available data sets there are versions of Bass and Paur (1985) cross-sections denoted as Brewer operational (Brw), IGACO quadratic coefficient (B&P), the cross-sections of Daumont Brion Malicet (DBM) (Daumont et al. (1992), Brion et al. (1993), and Malicet et al. (1995) ), and the newly recommended data set for ozone ground-based calculation by Serdyuchenco, Groshelev, Weber (SGW) (Serdyuchenko et al., 2012; Gorshelev et al., 2012; Weber et al., 2016).

Using the measured slit functions, the calculated effective ozone cross sections are 0.9% higher compared to those obtained by using the parametrized Brewer slits in the standard procedure (Table 2) independently of the cross sections used.





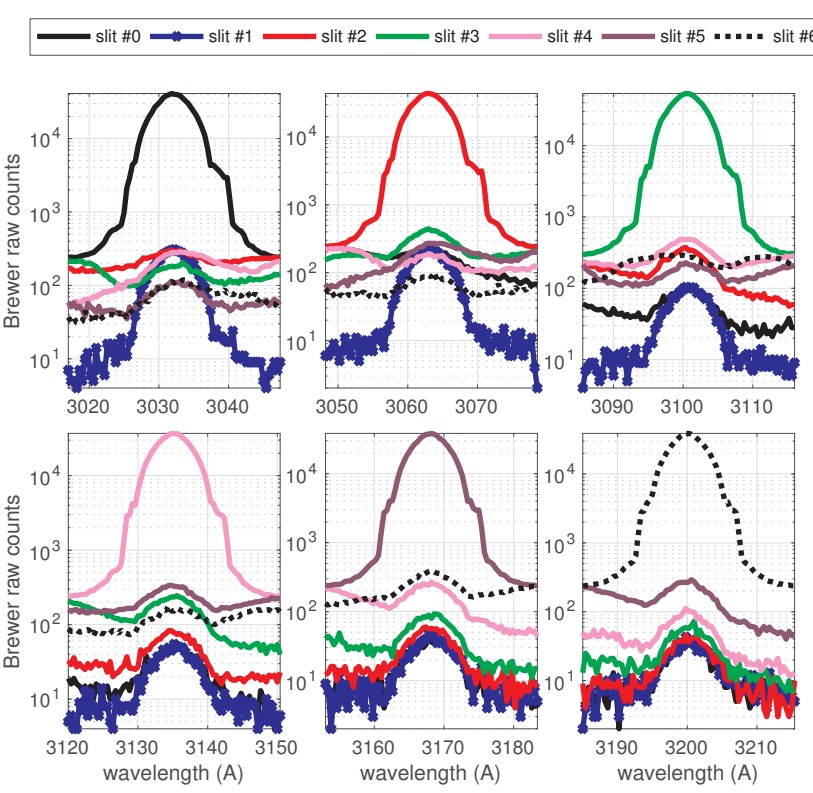

**Figure 5.** Brewer measurements in the ozone mode while the laser wavelength is changed every 0.04 nm. The blue curve corresponds to the dark counts obtained from the measurements of slit 1.

**Table 2.** Ozone absorption coefficient in atm $cm^{-1}$ calculated using four absorption cross sections.

|       | Parametrized | Measured |
|-------|--------------|----------|
| BRW   | 0.3381       | 0.3407   |
| B&P   | 0.3330       | 0.3360   |
| DMB   | 0.3483       | 0.3514   |
| SDK   | 0.3392       | 0.3422   |



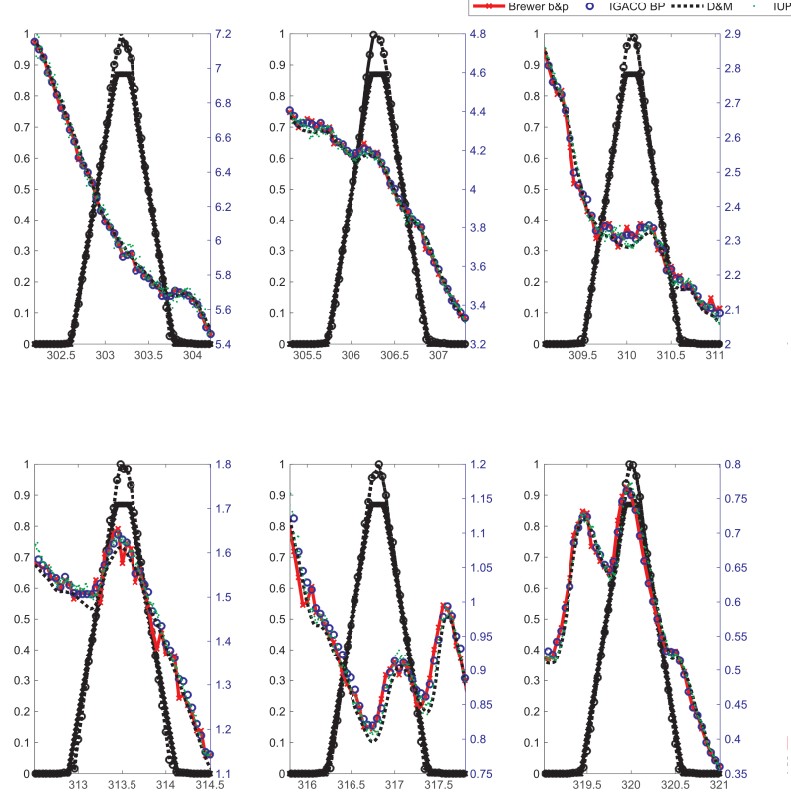

**Figure 6.** Plot of the parametrized (thick lines, left axis) and the measured slit functions (dots, left axis) as well as the different ozone cross sections in $cm^{-1}$ (right axis) used for the Brewer effective ozone absorption coefficient calculation.

## 4 Discussion

The experiment allows us to validate the Brewer standard methodology used to perform the wavelength calibration. For this purpose, we compare laser-based wavelength calibration results to those yielded by the standard operative method based on scanning the spectral lamps in case of both the quadratic and the cubic fit to the dispersion relation.

Figure 7 shows discrepancies between the central wavelengths calculated by the quadratic and the cubic fits to be bigger than 0.1Å for wavelength above 320 nm and much bigger near 350 nm. This is also indicated by a systematic shape of the residuals of the quadratic fit that are much larger than for the cubic fit (Figure 8). This indicates that the quadratic fit is only valid in the ozone range (310 nm - 320 nm) as already noted in Gröbner et al. (1998). The comparison of the calculated FWHMs (Figure 7 ) shows a different pattern with a difference of 0.1Å between the direct and the scanning methods in the ozone range and with a smaller difference between the quadratic and the cubic fits.





The differences to the directly calculated ozone absorption coefficients are summarized for the six measurements in table 3, taking as a reference the direct measurements. The quadratic fits result in bigger differences of around 1% whereas in the case of the cubic fits the differences decrease to 0.3% and 0.1% when the laser or the discharge lamps are used, respectively.

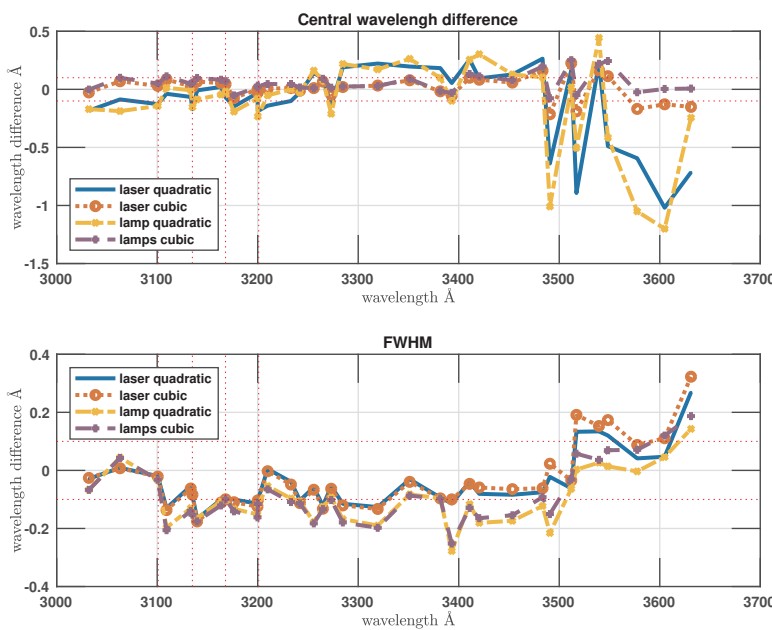

**Figure 7.** Differences between the central wavelengths (upper panel) and FWHMs (down panel) determinated directly and by the scanning methods: with the laser wavelengths in equally spaced grid every 5nm and lines of the discharge lamps; in both cases quadratic and cubic fitting are used.

## 5   Conclusions

1. Using the measured slit functions instead of the parametrized ones increases the ozone absorption coefficients and consequently the calculated ozone values by 0.8%.

2. The quadratic dispersion relation fit used in the standard Brewer algorithm is not suitable outside the ozone spectral range 310 nm - 320 nm. The residuals show a systematic pattern, which is particularly important in the higher end of the spectral range.

3. The comparison of the results of the three experiments shows a maximum difference of 0.3% if the cubic fit is used to approximate the dispersion relation of the Brewer instrument. The respective difference between the ozone absorption coefficient that is obtained from the direct measurements of the tunable laser in the ozone mode and from the opera-





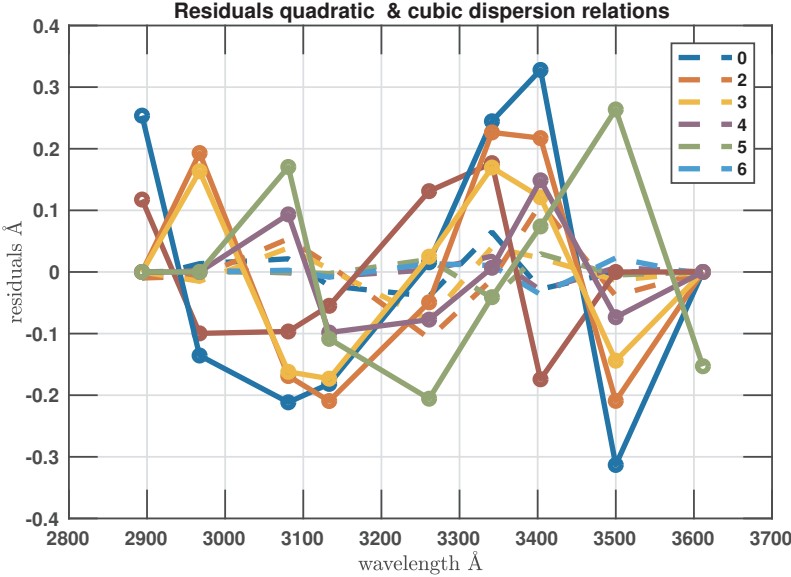

**Figure 8.** Residuals of the quadratic fit ( circles) and cubic (dashed) , the color indicates the six brewer slits with the laser wavelengths in equally spaced grid every 5nm

**Table 3.** Ozone absorption coefficient in atm $cm^{-}1$ calculated using four absorption cross sections

|  | brw_scan[1] | opo_quad[2] | opo_cubic[3] | lamp_quad[4] | lamp_cubic[5] |
|---|---|---|---|---|---|
| SGW | 0.3409 | 0.3442 | 0.342 | 0.3446 | 0.3412 |
| ratio | 1 | 0.9881 | 1.0033 | 1.0108 | 1.001 |

1. Laser wavelength scanned @ fixed Brewer grating position

2. Brewer grating position. changed @ fixed laser wavelength, dispersion approx. by a quadratic function

3. Brewer grating pos. changed @ fixed laser wavelength, dispersion approx. by a cubic function

4. Brewer grating pos. changed @ fixed lamp emission wavelength, dispersion approx. by a quadratic function

5. Brewer grating pos. changed @ fixed lamp emission wavelength, dispersion approx. by a cubic function

tive discharge lamp method is only of 0.1%, if both use the parametrized or measured slit. This confirms the standard procedure used for the RBCC-E calibrations.





*Acknowledgements.* This work has been supported by the European Metrology Research Programme (EMRP) within the joint research project ENV59 "Traceability for atmospheric total column ozone" (ATMOZ). The EMRP is jointly funded by the EMRP participating countries within EURAMET and the European Union.





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
