# Peer review of "Wavelength calibration of Brewer spectrophotometer using a tunable pulsed laser and implications to the Brewer ozone retrieval"

_Atmospheric Measurement Techniques, 2017_

## Referee Comment (RC1) · J. Gröbner (Referee) · 15 Feb 2018

The paper describes a very detailed and unique characterisation experiment of a Brewer spectrophotometer to determine its spectral characteristics (wavelength scale and spectral resolution) which is necessary to calculate the ozone absorption coefficient required for the total column ozone determination from the solar irradiance measurements. The experiment was performed using a tunable laser source to compare and validate the standard procedure used by the Brewer community. The results show that the two procedures provide consistent results to within 0.1% which is very satisfying and confirms that the current standard procedure is valid.

[Figure]

Apart from minor grammatical errors the manuscript is well written and certainly interesting to the scientific community. I have a few comments which I would the authors to answer, pending those I support the publication of the manuscript.

- page 6, line 10: It is a cubic polynomial fit, not cubic spline.

- page 7, lines 13-17: It is true that the hg test of the Brewer is repeated when the discrepancy between the actual and determined position is larger than 1.5 steps. However the hg routine sets the position of the micrometer according to the calculation, and repeating the hg routine serves mainly for confirmation. Therefore the hg routine is accurate to +-0.5 steps, since this is the resolution of the system, not +-1.5 steps as written in the manuscript. This considerably improves the estimated wavelength uncertainty.

- In section 3.1, I would suggest to add some information on the wavelength uncertainty of the tunable laser setup, which will affect the Brewer wavelength dispersion. I expect in fact the Brewer wavelength dispersion to have less uncertainties when using spectral discharge lamps with published emission line wavelengths ($\sim$1 pm), than the wavelength obtained by the tunable laser system ($\sim$10 pm).

- In section 3.1.2 the authors compare the ozone absorption coefficient calculated with the parametrized and the actual slit functions and show that the difference is of the order of 0.9% (Table 2). The parametrized slits however are trapezoidal, with a plateau at 0.87 (13% from the top). However as shown in Figure 6, this is not representing the true slits, and therefore the parametrization might be closer to reality when using the full triangle as parametrization. This might show that the method using the parametrization with a full triangle will have less differences to the tunable laser results using the actual measured slit functions. (I have made some tests and the full triangle parametrization resolves about half of the 0.9% discrepancy). I would suggest that the authors add a third column in table 2 showing this information.

- In the conclusion, page 13, last sentence, I do not understand the statement saying

that both methods agree to 0.1% if the parametrized or measured slits are used. To my understanding, the standard method using a scanning grating is not able to use the measured slits, since the method relies on interpolating the slit functions to the ozone position, which therefore requires a parametrized slit.

- In my opinion the abstract should also mention the positive result that the tunable laser and the scanning grating method give the same ozone absorption coefficients (to within 0.3% or so)? Minor comments:

- The different wavelength scales (nanometer, angstroem) used in the manuscript and the figures is confusing, and I would recommend to use a single one (nanometer)?

-page 4, line 8 : i would explicitly state that the method is an ozone calibration (not to be confused with a radiometric irradiance calibration for example).

page 5, point 3: The FWHM also depends on wavelength, which therefore requires some sort of parametrization of the slit function when using the standard scanning grating method.

- Figure 2: The units on the left axis seem too small (maximum of 7 counts/second)?

- page 6, line 3: I would remove the value in parenthesis (0.0080 nm), or replace the picometer values. - Table 3, I did not find the acronym for SGW. Could it be added in the caption of the Table, for clarity?

---

## Referee Comment (RC2) · L. Doppler (Referee) · 28 Feb 2018

General comment

The paper describes an experiment of a spectral characterization of a meteorological instrument: The Brewer (principally used for UV radiation and total ozone column monitoring). This experimentation has been done in a laboratory of a metrological institution (the German PTB in Braunschweig) with the help of one of the most modern and precise material: A tunable laser. The originality of this experiment (part of the EMRP/ATMOZ project), is that scientific teams of the meteorological community worked together with metrology specialists, what is a unique gain for the quality man-

agement of the meteorological measurements. The spectral characterization that has been done is necessary for the determination of the total ozone column from the UV irradiance measured with the instrument Brewer.

The results of the experiment allow a high quality validation of the standard procedures of spectral characterization used in the Brewer community, and allow defining quantitatively the limits of these standard procedures and of their assumptions.

The results are well presented in this article, and the discussion is of high scientific quality and will be useful for the Brewer community.

Apart from minor (but many) typology and grammatical errors, that should be corrected (I have listed a part of them at the end of this report), the manuscript is well written pleasant to read and well understandable. I have a few (5) comments/questions which I would the authors to answer, pending those I support the publication of the manuscript.

Technical comments ("x-y = page x, Line y):

5 Comments/questions about the content:

1) In the beginning of the background paragraph, when you mention the slits of the "slit mask" (2-1), it would be welcome to have a brief description of the brewer's slits, and to what they are useful. Maybe you should introduce a table of them and refer to the table in the text.

2) (in 1. Background), lines 2-10 to 2-15, you mention some assumptions about the slit functions (2-14). It would be nice to explain before or in a table, what are the most important assumptions that are done.

3) In the description of the experiment (Background from 2-19 to 2-28):

3a) I guess the use of the tunable laser is useful for point 2, not point 3. So the mention "using the tunable laser" (2-24) should go in the title of point 2 (2-20).

3b) In point 3, you mention that the Brewer scans +-2nm around a fixed wavelength of

the laser. Further in Point 3, you mention that the Brewer can scan with 5nm increment between 290 nm and 365 nm. You need to precise how you can scan with a lower step than the increment.

4) In the alpha-formula (4-24) you write the ozone cross-section with alpha(lambda) and then in the text, you mention sigma for it.

5) In the discussion, it would be welcome to explain how the different Brewer users can use these results to optimize the TOC retrieval with their own Brewer. Can these results be generalized to all Brewers? To all Mk-III Brewers? Or should every Brewer go to a laboratory with a tunable laser to characterize its slits? Do you have assumption concerning the age stability of this slit characterization? Is it expected to change with the time? If yes why?

Grammar and typos:

You use sometimes "tuneable", sometimes "tunable", please choose one terminology, and I guess "tunable" is the correct one.

-> So please correct in title, in 1-5 (Abstract), in 2-9, 2-13 and 2-16 (Chapter 1), in 7-22 and 8-1 (Chapter 3)

Abstract:

1-9: "a underestimation" -> an underestimation

1. Background

1-13: "total column ozone (TOC)"-> total ozone column (TOC)

2-9 "the use of the use of" -> the use of

2-9 "allow us" -> allows us

2-10 "coefcients" -> coefficients

2-14 "the need for the assumptions" -> the need of assumption

2. Calibration of the Brewer sprectrophotometer

3-7 "weighting coefficients w" -> "weighting coefficients wi"

5-11 "referered" -> referred

5-12 "double Brewer ," -> double Brewer, (without space before the ",")

6-1 "one of the six exit slit" -> one of the six exit slits

6-7 "0.7 A" -> 0.7 Å (A with "o" on the top for Ångström)

7-1 "The cubic approximation method. . . use knowledge" -> ". . . uses . . ."

3. Pulsed laser-based measurements

8-10 and 8-11 "counts/seconds" (3 times)-> counts/second

9-7 "parametrised" -> parametrized

9-8 "( Orphal et al. (2016)." -> ", Orphal et al. (2016)." or (Orphal et al. (2016))

4. Discussion

12-1 "in table 3" -> in table 3

Table 3, legend: please replace "@" with "at"

[Figure]

---

## Author Comment (AC1) · 10 May 2018

**Wavelength calibration of Brewer spectrophotometer using a tunable pulsed laser and implications to the Brewer ozone retrieval**

Alberto Redondas1,2, Saulius Nevas3, Alberto Berjón4,2, Meelis-Mait Sildoja3, Sergio F. León-Luis1,2, Virgilio Carreño1,2, and Daniel Santana4,2

1Agencia Estatal de Meteorología, Izaña Atmospheric Research Center, Spain
 2Regional Brewer Calibration Center for Europe, Izaña Atmospheric Research Center, Tenerife, Spain
 3Physikalisch-Technische Bundesanstalt (PTB), Braunschweig, Germany
 4University of La Laguna, Department of Industrial Engineering, S.C. de Tenerife, Spain

Correspondence to: Alberto Redondas (aredondasm@aemet.es)

We thank the referee for their detailed comments on our manuscript. In the attached document we address their questions and suggestions.

**Comments by Reviewer #1**

J. Gröbner (Referee) julian.groebner@pmodwrc.ch The paper describes a very detailed and unique characterisation experiment

- 5 of a Brewer spectrophotometer to determine its spectral characteristics (wavelength scale and spectral resolution) which is necessary to calculate the ozone absorption coefficient required for the total column ozone determination from the solar irradiance mea- surements. The experiment was performed using a tunable laser source to compare and validate the standard procedure used by the Brewer community. The results show that the two procedures provide consistent results to within 0.1% which is very satisfying and confirms that the current standard procedure is valid. Apart from minor grammatical errors the
- 10 manuscript is well written and certainly inter- esting to the scientific community. I have a few comments which I would the authors to answer, pending those I support the publication of the manuscript.

page 6, line 10: It is a cubic polynomial fit, not cubic spline.

**corrected**

- page 7, lines 13-17: It is true that the hg test of the Brewer is repeated when the discrepancy between the actual and determined position is larger than 1.5 steps. How- ever the hg routine sets the position of the micrometer according to the calculation, and repeating the hg routine serves mainly for confirmation. Therefore the hg routine is accurate to +-0.5 steps, since this is the resolution of the system, not +-1.5 steps as written in the manuscript. This considerably improves the estimated wavelength uncertainty. Yes this reduces considerably the error, 1.5 steps are an estimation of the maximum error in the EUBREWNET processing, in this network a mercury test is considered failed when is repeated, the ozone observations are discarded between the last test and the failed one, how ever most of the test gives a correction of less than step.

In section 3.1, I would suggest to add some information on the wavelength uncertainty of the tunable laser setup, which will affect the Brewer wavelength dispersion. I expect in fact the Brewer wavelength dispersion to have less uncertainties when using spectral discharge lamps with published emission line wavelengths ( $\approx 1pm$ ), than the wavelength obtained by the tunable laser system ( $\tilde{1}0$  pm).

The text in Section 3.1 was expanded to explain the wavelength calibrations of the instruments used to monitor the laser wavelengths. As to the uncertainty of the Brewer calibration using emission lines, one has to consider also the effect of varying ambient conditions (pressure, temperature, humidity, CO2 content) on the refractive index of air, which is also in the range of several picometers (e.g. ambient temperature change from 20 C to 25 C has an effect of 2 pm, atmospheric pressure change from 101kPa to 98 kPa would cause a 3pm change). So that the uncertainties in both cases should be indeed quite comparable.

- In section 3.1.2 the authors compare the ozone absorption coefficient calculated with the parametrized and the actual slit functions and show that the difference is of the order of 0.9% (Table 2). The parametrized slits however are trapezoidal, with a plateau at 0.87 (13% from the top). However as shown in Figure 6, this is not representing the true slits, and therefore the parametrization might be closer to reality when using the full triangle as parametrization. This might show that the method using the parametrization with a full triangle will have less differences to the tunable laser results using the actual measured slit functions. (I have made some tests and the full triangle parametrization resolves about half of the 0.9% discrepancy). I would suggest that the authors add a third column in table 2 showing this information.

10

Thanks for the suggestion, a new column were added and the calculation repeated, we found the discrepancy is more near to 0.85% +/- 0.1 for the high resolution cross sections rather than 0.9% and the use of the triangular parametrization in fact reduce the difference to the half 0.45%. The results slightly differ from cross section to cross section with the exception of t Bass&Paur probably due his low spectral resolution (Table 1).

2

Table 1. Ozone absorption coefficient in atm  $cm^{-1}$  calculated using four absorption cross sections. And the percentage difference to the measured to the paramerizated with a trapezoid and triangular

| Trapezoid | Triangular | Measured | % Trapezoid | % Triangular |
|-----------|------------|----------|-------------|--------------|
| 0.3381    | 0.3395     | 0.3406   | -0.73       | -0.33        |
| 0.3331    | 0.3344     | 0.3359   | -0.85       | -0.44        |
| 0.3483    | 0.3498     | 0.3514   | -0.86       | -0.45        |
| 0.3393    | 0.3407     | 0.3422   | -0.84       | -0.43        |

- In the conclusion, page 13, last sentence, I do not understand the statement saying that both methods agree to 0.1% if the parametrized or measured slits are used. To my understanding, the standard method using a scanning grating is not able to use the measured slits, since the method relies on interpolating the slit functions to the ozone position, which therefore requires a parametrized slit.

- In my opinion the abstract should also mention the positive result that the tunable laser and the scanning grating method give the same ozone absorption coefficients (to within 0.3% or so)?

**The comment were added**

**1** Minor comments**

**5 :**

- The different wavelength scales (nanometer, angstroem) used in the manuscript and the figures is confusing, and I would recommend to use a single one (nanometer)?

Following your suggestion, we have modified the Figures and, now, its scale is in nanometers. In the text we have tried to express all the results in nanometers. However, some results were expressed as Angstroem or picomentros, because it is a very small value. (Also, the other referee has indicated that some results may be better expressed in picometers) -page 4, line 8 : i would explicitly state that the method is an ozone calibration (not to be confused with a radiometric irradiance calibration for example).

**revised.**

page 5, point 3: The FWHM also depends on wavelength, which therefore requires some sort of parametrization of the slit function when using the standard scanning grating method.

revised.

```
- Figure 2: The units on the left axis seem too small (maximum of 7 counts/second)?
```

5

The exponential were missing, now is corrected.

```
- page 6, line 3: I would remove the value in parenthesis (0.0080 nm), or replace the picometer values.
```

The value has been removed in the text.

-Table 3, I did not find the acronym for SGW. Could it be added in the caption of the Table, for clarity?

10 The SGW makes reference to the article Weber et al. (2016). In this paper, the authors studied the ozone absorption cross section and its uncertainty.

**2 initial review**

Page 1, Background: Some references on the brewer spectrophotometer should be included.

References to the brewer instrumet are added.

I have never seen such an equation to determine the ozone absorption coefficient within the beer-Lambert law. I even think that it is a circular argument, because the solar spectrum would be included twice, one time directly in the Beer-Lambert Law, the second time for the convolution. Therefore I think this equation is in fact wrong, and the correct one is the one mentioned later (e.g. equation 8. I would suggest that either the authors can provide a reference for this usage of Equation 7, or delete this part of the discussion, which is not used anyway.

Simmilar equation were used in a tutorial by Davd Wardle during the Brewer workshop,

Wardle, D.: Physical Principles II: Optical Characteristics of the Brewer, in The Tenth Biennial WMO Consultation on Brewer Ozone and UV Spectrophotometer Operation, Calibration and Data Reporting, World Meteorological Organization. [online]

5 Available from: "ftp://ftp.wmo.int/Documents/PublicWeb/arep/gaw/gaw176\_10thbrewer.pdf" 2008

and on the Thesis Savastiouk (2005) page 94, eq 4.12

Is a simplified reformulation of the more complex equation of Vernier-Wardle (Vanier and Wardle, 1969) used on the Dobson by Bernhard et al. (2005) (Eq 7)

$$\alpha(\lambda) = \frac{\log\left(\frac{\int E_o(\lambda)S(\lambda,\lambda_i)10^{-\alpha(\lambda)X\mu-\beta(\lambda)}\frac{P}{P_o}\nu\,d\lambda}{\int E_o(\lambda)S(\lambda,\lambda_i)10^{-\beta(\lambda)}\frac{P}{P_o}\nu\,d\lambda}\right)}{X\mu} \tag{1}$$

10 This references were added to the article:

Page 6 line 10, Figure number is missing

**corrected**

Page 7, lines 1-8: I think there is a mistake in the discussion in the precision of the hg test: The HG test allows the wavelength setting to be set to within 0.5 micrometer steps, not 1.5. Then, since one step equals (approximately), 0.0075 nm, 0.5 steps would correspond to 0.00375 nm, or 0.0375A.

Yes there was a mistake, it should be 0.866 steps not Å. The Hg test allows indeed the wavelength setting to be set within 0.5
micrometer step. However, the correction is only applied if the difference is greater than 2 steps. According to the Brewer SOP :
"Corrections to the micrometer position are made, and if the adjustment required is greater than 2 steps (.012nm) then the scan is repeated. "So that we assume that the maximum error is +-1.5 steps with a rectangular probability distribution function. Hence, the standard uncertainty in such a case is 1.5/sqrt(3) = 0.866 steps, which is approx 0.065Å (0.0075 nm per step)

Figure 5. Why use of log scale? In this manuscript, the interest lies in the slit function shape, and therefore in the linear scale, not in the wings.

The reason to use log scale is to visualize the dark count issue, in linear scale is difficult to see.

page 10, last line. I would add "as already noted in Gröbner et al., 1998", at the end of the sentence finishing with (310-320 nm).

**5 added.**

I am confused by the conclusion, and would suggest that the authors rephrase it be less ambiguous: Point 1 of the conclusion states that the tunable laser results give an ozone absorption coefficient 0.8% higher than the ones obtained with the standard approach using the parametrized slits. The last point states however that the use of the cubic spline agrees with the tunable laser results to within 0.1% and therefore confirms that the use of the tunable laser (fixed grating position), with the standard procedure (rotating grating) is equivalent. While I understand that cubic and quadratic dispersion fittings might give rise to differences, I do not understand how this affects the use of parametrized slits and actually measured ones using a tunable laser.

We rephrase the conclusions, to reflect from one side the use of parametrized slit function gives a significant difference of 0.8% if we use the measured slit, but the use of the laser confirm the operational rotating grating method (using the parametrized slits with the laser)

**References**

- Bernhard, G., Evans, R. D., Labow, G. J., and Oltmans, S. J.: Bias in Dobson total ozone measurements at high latitudes due to approximations in calculations of ozone absorption coefficients and air mass, Journal of Geophysical Research: Atmospheres, 110, n/a–n/a,
- 5 https://doi.org/10.1029/2004JD005559, http://dx.doi.org/10.1029/2004JD005559, d10305, 2005. Savastiouk, V.: Improvements to the direct-sun ozone observations taken with the Brewer spectrophotometer, 2005.

Vanier, J. and Wardle, D.: The effects of spectral resolution on total ozone measurements, Quarterly Journal of the Royal Meteorological Society, 95, 399, 395, http://adsabs.harvard.edu/abs/1969QJRMS..95..395V, 1969.

Weber, M., Gorshelev, V., and Serdyuchenko, A.: Uncertainty budgets of major ozone absorption cross sections used in UV remote sensing ap-

10 plications, Atmospheric Measurement Techniques, 9, 4459–4470, https://doi.org/10.5194/amt-9-4459-2016, https://www.atmos-meas-tech. net/9/4459/2016/, 2016.

---

## Author Comment (AC2) · 10 May 2018

**Wavelength calibration of Brewer spectrophotometer using a tunable pulsed laser and implications to the Brewer ozone retrieval**

Alberto Redondas[1,2], Saulius Nevas[3], Alberto Berjón[4,2], Meelis-Mait Sildoja[3], Sergio F. León-Luis[1,2], Virgilio Carreño[1,2], and Daniel Santana[4,2]

[1]Agencia Estatal de Meteorología, Izaña Atmospheric Research Center, Spain
[2]Regional Brewer Calibration Center for Europe, Izaña Atmospheric Research Center, Tenerife, Spain
[3]Physikalisch-Technische Bundesanstalt (PTB), Braunschweig, Germany
[4]University of La Laguna, Department of Industrial Engineering, S.C. de Tenerife, Spain

*Correspondence to:* Alberto Redondas (aredondasm@aemet.es)

We thank the referee for their detailed comments on our manuscript. In the attached document we address their questions and suggestions.

**Comments by Reviewer #2**

**1 General comment**

5   The paper describes an experiment of a spectral characterization of a meteorological instrument: The Brewer (principally used for UV radiation and total ozone column monitoring). This experimentation has been done in a laboratory of a metrological institution (the German PTB in Braunschweig) with the help of one of the most modern and precise material: A tunable laser. The originality of this experiment (part of the EMRP/ATMOZ project), is that scientific teams of the meteorological community worked together with metrology specialists, what is a unique gain for the quality management of the meteorological

10   measurements. The spectral characterization that has been done is necessary for the determination of the total ozone column from the UV irradiance measured with the instrument Brewer. The results of the experiment allow a high quality validation of the standard procedures of spectral characterization used in the Brewer community, and allow defining quantitatively the limits of these standard procedures and of their assumptions. The results are well presented in this article, and the discussion is of high scientific quality and will be useful for the Brewer community.

15   Apart from minor (but many) typology and grammatical errors, that should be corrected (I have listed a part of them at the end of this report), the manuscript is well written pleasant to read and well understandable. I have a few (5) comments/questions which I would the authors to answer, pending those I support the publication of the manuscript. Technical comments ("x-y = page x, Line y):

5 Comments/questions about the content:

> 1) In the beginning of the background paragraph, when you mention the slits
> of the "slit mask" (2-1), it would be welcome to have a brief description of
> the brewer's slits, and to what they are useful. Maybe you should introduce a
> table of them and refer to the table in the text.

A table with the slits (Number and its wavelength) and a description has been added in the text.

> 2) (in 1. Background), lines 2-10 to 2-15, you mention some assumptions about
> the slit functions (2-14). It would be nice to explain before or in a table,
> what are the most important assumptions that are done.

The assumptions are described later, we add a reference to the section on the text.

> 3) In the description of the experiment (Background from 2-19 to 2-28):
> 3a) I guess the use of the tunable laser is useful for point 2, not point
> 3. So the mention "using the tunable laser" (2-24) should go in the title of
> point 2 (2-20). yes 3b) In point 3, you mention that the Brewer scans +-2nm
> around a fixed wavelength of the laser. Further in Point 3, you mention that
> the Brewer can scan with 5nm increment between 290 nm and 365 nm. You need to
> precise how you can scan with a lower step than the increment.

Yes is an error, the experiment were performed with the laser emitting from 290 to 355 every 5 nm whereas the brewer is scanning every 5 steps.

> 4) In the alpha-formula (4-24) you write the ozone cross-section with
> alpha(lambda) and then in the text, you mention sigma for it.

Corrected

> 5) In the discussion, it would be welcome to explain how the different Brewer
> users can use these results to optimize the TOC retrieval with their own
> Brewer. Can these results be generalized to all Brewers? To all Mk-III Brewers?
> Or should every Brewer go to a laboratory with a tunable laser to characterize
> its slits? Do you have assumption concerning the age stability of this slit
> characterization? Is it expected to change with the time? If yes why?

The work validate the method currently in use with the brewer network, the limitations of the quadratic dispersion is an issue on MK-III and MK-IV brewer and the new operating software recently introduce the cubic dispersion.We don't think necessary the characterization with the laser of network brewer.

We introduce this on the discussion.

**5 Grammar and typos**

You use sometimes "tuneable", sometimes "tunable", please choose one terminology, and I guess "tunable" is the correct one.
-> So please correct in title, in 1-5 (Abstract), in 2-9, 2-13 and 2-16 (Chapter 1), in 7-22 and 8-1 (Chapter 3) **Done**

**Abstract**

1-9: "a underestimation" -> an underestimation **Done**

**1.Background**

1-13: "total column ozone (TOC)"-> total ozone column (TOC) 2-9 "the use of the use of" -> the use of 2-9 "allow us" -> allows us **Done**

2-10 "coefcients" -> coefficients **Done**

2-14 "the need for the assumptions" -> the need of assumption **Done**

**2. Calibration of the Brewer sprectrophotometer 3-7 "weighting coefficients w" -> "weighting coefficients wi" **Done**

5-11 "referered" -> referred **Done**

5-12 "double Brewer ," -> double Brewer, (without space before the ",") **Done**

6-1 "one of the six exit slit" -> one of the six exit slits **Done**

6-7 "0.7 A" -> 0.7 Å (A with "o" on the top for Ångström) **Done**

7-1 "The cubic approximation method. . . use knowledge" -> ". . . uses . . ." **Done**

**3. Pulsed laser-based measurements**

8-10 and 8-11 "counts/seconds" (3 times)-> counts/second **Done**

9-7 "parametrised" -> parametrized **Done**

9-8 "( Orphal et al. (2016)." -> ", Orphal et al. (2016)." or (Orphal et al. (2016)) **Done**

**References**

---

## Author Response (AR1)

**Wavelength calibration of Brewer spectrophotometer using a [..\*] tuneable pulsed laser and implications to the Brewer ozone retrieval**

Alberto Redondas1,2, Saulius Nevas3, Alberto Berjón4,2, Meelis-Mait Sildoja3, Sergio F. León-Luis1,2, Virgilio Carreño1,2, and Daniel Santana4,2

1Agencia Estatal de Meteorología, Izaña Atmospheric Research Center, Spain

[revised manuscript text omitted]

 $[..^{80}]$

The cubic method of Gröbner et al. [..81] use the optical design of the [..82] brewer to transfer line measures from one slit to the other slits [..83] reducing the number of free parameters to [..84] adjust compared with the quadratic method. However

15 [..85] there is a systematic difference [..86] on the calculation of the ozone absorption coefficient between the two methods.
 Both methods generally agree [..87] if only the brewer ozone spectral range [..88] is used (Redondas and Rodriguez-Franco, 2012).

The stability of the wavelength calibration during Brewer operations is checked by measuring the internal Hg lamp. In most of the Brewers, the 302 nm double line (302.150 nm and 302.347 nm) is used due to its proximity to the Brewer operational wavelengths. However, for Brewer #185 and for an increasing number of other Brewers the test is performed using the more powerfull 296.7 nm line. The wavelength test includes 12 measurements of the line from the mercury lamp on slit 0, with an

<sup>75removed: (Gröbner et al., 1998)

<sup>76removed: Slit function of the slit #3 (320 nm) measured in the scanning mode: the standard method uses the normalized values only between 0.2 and 0.8 (points with crosses). The figure shows the signals recorded during the scans (blue diamonds) and the non-linearity-corrected values (black circles) in counts/second. The hysteresis is evident from the asymmetry of the uncorrected signals in the low signal region of the plot. The center steps and the FWHMs calculated from the up-scan are shown in red while the values derived from the down-scanned slit function are presented in blue text. The central wavelength determination is not affected by the nonlinearity but the apparent FWHMs would be larger if the non-linearity correction were not applied. The resulting parametrized slits are represented on the right axis.

<sup>77removed: Å

<sup>78removed: 2

<sup>79removed: cubic polynomial

<sup>80removed: The cubic approximation

81 removed: uses knowledge of

<sup>82removed: Brewer spectrometer to transfer results of the spectral line measurements

83 removed: , which reduces

<sup>84removed: be adjusted

<sup>85removed:,

<sup>86removed: between the ozone absorption coefficients calculated by

<sup>87removed: only in the

<sup>88removed: of the Brewer instrument

Figure 2. Measured slit function of the 320 nm (slit #3): the method uses the normalised values only between 0.2 and 0.8 (point with cross). The calculated center step and the FWHM are shown for the up (red) and down scan (blue). The resulting parametrized slit are represented on the right axis. Uncorrected counts/second (blue diamonts) and non-linearly-corrected values (black circles). The central wavelength determination is not affected by the nonlinearity but the FWHM is bigger if the correction is not applied. The hysteresis is evident from the asymmetry of the function at the low values region of the plot.

increment of  $\pm 10$  steps of the micrometer motor. The obtained curve for the line peak is compared to a stored reference one. The comparison is done by shifting the two [..89]scaned curves against each other and calculating the correlation coefficient between the two after each shift. The interpolated step number yielding the maximum of the correlation coefficient provides the reference micrometer position (Savastiouk, 2005). If the required adjustment of the micrometer position is more than one

5

and a half motor steps, the test is repeated. [..90] The accuracy of the wavelength setting of the Brewer instrument achieved by such an approach is [..91] limited by one and half motor steps and cannot be better than about 0.86A. This affects the ozone [..92] abortion coefficient by approximately of [..93] 0.10 atmcm-1 and the ozone concentration by 0.3%.

**3 Pulsed laser-based measurements**

**3.1 Instrumental setup**

<sup>89removed: scanned

90 removed: Hence, the

 $^{91}$ removed: defined by a rectangular probability distribution function on the interval of +-0.5 steps. Thus, the respective standard uncertainty is 0.5/sqrt(3)= 0.289 steps. During the brewer operation at RBCC-E calibration campaigns the ozone observations are discarded if the subsequent hg test is repeated, the bigger discrepancy for the accepted measurements will be 1.5 steps. This 1.5

92 removed: absorption

93 removed: 0.1